# Effects of soccer instruction on the executive functions and agility of children in early childhood

**Sho Aoyama** *

Department of Education, Yamaguchi University, Yamaguchi City, Yamaguchi Prefecture, Japan

* aoyama@yamaguchi-u.ac.jp

## Abstract

Studies have shown that in open-skill sports the executive function of high-performing players is significantly higher than that of the control group. However, whether participation in soccer effectively improves executive function is unclear because previous studies lacked baseline measurements of executive function. Furthermore, agility, mostly developed in early childhood, is the most demanding component of physical fitness in open-skill sports, requiring sudden changes in body direction while running at full speed. However, no studies have examined the improvement in agility of young children participating in open-skill sports through comparison with a control group. This study aimed to clarify whether instruction in soccer, an open-skill sport, can effectively improve the executive function and agility of kindergarteners compared to a control group. In April 2020, 31 five-year-old children enrolled in kindergarten K in H prefecture in Japan were targeted as the intervention group and received soccer instruction for 12 weeks. In April 2020, a control group was established, consisting of 39 five-year-old children enrolled at the same kindergarten. Participants in both groups were measured for agility, inhibition, and working memory before and after (pre–post) soccer instruction for the intervention group. For each item, a two-way ANOVA of the group (intervention group control group) x measurement period (pre–post) was performed, showing no significant interactions for working memory and agility. Because only the inhibition effect was significant, simple main effects were tested. Regarding inhibition, although the intervention group ($M = 0.18$, $SD = 0.01$) and the control group ($M = 0.17$, $SD = 0.01$) did not differ significantly in performance pre-intervention, the intervention group ($M = 0.23$, $SD = 0.01$) showed significantly better performance post-intervention compared to the control group ($M = 0.19$, $SD = 0.01$) ($p < .01$). Thus, soccer instruction improved kindergarteners' inhibition, to a level significantly higher than that of the control group. Still, no differences were found between groups regarding working memory and agility.

## Introduction

Exercise is important to improve physical fitness. In recent years, research suggesting that exercise is effective not only for improving physical fitness but also cognitive function, known

**Data Availability Statement:** All relevant data are within the paper and its Supporting Information files.

**Funding:** This work was supported by a Grant-in-Aid for Early-Career Scientists Research from the

Japan Society for the Promotion of Science [Grant Number 24K20638]. The funders had no role in study design, data collection and analysis, decision to publish, or preparation of the manuscript.

as executive function, has been attracting attention. Executive function is the ability to achieve goals by engaging in higher-order cognitive and behavioral control and is composed of three components: inhibition, working memory, and switching [1]. Inhibition is the ability to control dominant and automatic responses according to context [1]. Working memory is the ability to store and process information temporarily and use the stored information [1]. Switching is the ability to change the perspective of attention and prepare cognitively for the next task rule [1]. Early childhood is an ideal time to provide interventions intended to improve executive function, as executive function develops considerably during this period [2]. In addition, executive function plays a crucial role in children's future health status and socioeconomic status as well as school readiness in terms of behavior and learning [3–6]. Therefore, improving executive function during early childhood is critical.

Studies have suggested that aerobic exercise is effective in improving executive function in children [7]. For example, in a study of children with an average age of 9.6 years, performance on a task measuring working memory was compared between two groups, one with high aerobic capacity and the other with low aerobic capacity, based on the number of laps in a 20-meter shuttle run, which measures aerobic capacity. The results showed that the working memory of the group with high aerobic exercise capacity was significantly higher than that of the group with low aerobic exercise capacity [8]. In a study of children aged 8 to 12 years, children engaged in 12 weeks of aerobic exercise with increasing workload over time showed significantly better inhibition performance than children in a group engaged in a standard physical education class [9]. Davis et al. [10] reported that a 13-week aerobic exercise-based program for 7- to 11-year-old children who were inactive and overweight resulted in substantially improved inhibition compared to the control group.

On the other hand, in recent years, studies have indicated that performing complex exercises with high cognitive demands is more effective in improving executive function than performing simple exercises such as aerobic exercises [11]. Therefore, engagement in open-skill sports, such as soccer and tennis, defined as sports in which the sports environment is constantly changing and unpredictable, such as complex exercises with high cognitive demands, may be effective in improving executive function. Wang et al. [12] compared the performance of a task measuring inhibition between tennis players in open-skill sports, swimmers in closed-skill sports, and non-athletes as a control group. As a result, tennis players performed significantly better than swimmers and non-athletes in the task measuring inhibition. It has been reported that the performance of elite child soccer players in the task measuring inhibition was significantly higher than that of non-elite soccer players [13, 14]. In addition, a study of elite child soccer players revealed that players with higher working memory capacity had higher motor skills required for soccer [15].

Several studies have shown that in open-skill sports, particularly soccer, the performance of high-performing players on tasks measuring executive functions, such as inhibition and working memory, was significantly higher than that of the control group. However, the lack of baseline measurements is a major limitation in these previous studies. That is, because the original level of executive function prior to soccer participation is unknown, whether soccer participation is effective in improving executive function remains unclear. Establishing a causal relationship between soccer participation and improved executive function after baseline measurements is a task for future research. Due to the constant changes in the surrounding environment during a game, soccer requires players to keep abreast of constantly changing information about their surroundings and select appropriate behavior while inhibiting their pre-planned responses in response to changing circumstances [16, 17]. Therefore, participation in soccer is expected to improve inhibition and working memory abilities, which are components of executive function.

Furthermore, in open-skill sports, athletes' physical fitness is crucial for high performance in games [18]. Among the elements of physical fitness, agility, especially related to quick whole-body movements to change speed and direction in response to external stimuli, is the most important element, requiring sudden changes in body direction while running at full speed [19, 20]. Although it is not possible to ascertain the level of agility before engaging in open-skill sports, high levels of agility have been observed in athletes of various open-skill sports [21]. Agility is required for instantaneous responses to the constantly changing external environment of opponents, teammates, and the ball [20] and is thought to be enhanced through participation in open-skill sports. In addition, agility, which is closely related to nervous system development, is a crucial component of physical fitness that contributes considerably to motor skills in early childhood [22]. However, to date, no studies have examined the improvement in agility of young children participating in open-skill sports through comparison with a control group.

## The present study

Does soccer instruction improve executive function and agility, which are markedly developed in early childhood? This study aimed to clarify, through comparison with a control group, whether soccer instruction can effectively improve executive function and agility in inexperienced open-skill sports participants in early childhood. By targeting inexperienced open-skill athletes, it was possible to examine whether engaging in open-skill sports increased executive function and agility, considering the differences in original executive function and agility not been revealed in previous studies. This study hypothesized that the intervention group that received soccer instruction would show significantly improved executive function and agility compared to the control group. This study is expected to create new value for participation in open-skill sports by effectively enhancing executive function, an important ability for school readiness in terms of behavior and learning, and agility, an important component of physical fitness, in early childhood.

## Materials and methods

### Participants

The intervention group consisted of 31 five-year-old children (17 boys, 14 girls; mean age: 71.21 ± 0.5 months) enrolled at K kindergarten in H prefecture, Japan, in April 2020. The control group consisted of 39 five-year-old children (18 boys, 21 girls; mean age: 72.4 ± 0.6 months) enrolled at the same K kindergarten in H prefecture as the intervention group, in April 2020. When recruiting participants, individuals with prior participation in open-skill sports, including soccer, were excluded. Therefore, all participants were children who had never previously participated in open-skill sports, including soccer. Sample size calculations with G*Power 3.1 indicated that 31 children in the intervention group and 39 children in the control group had a power $(1-\beta)$ of 0.80, effect size of 0.15, the number of measurements = 3, correlation among repeated measures = 0.5, nonsphericity correction e = 1, and an $\alpha$ level of .05 [23].

### Procedure

The intervention group received soccer instruction once a week for 30 minutes in the morning for 12 weeks, from May to July 2020, in a field next to their kindergarten building. Specifically, as an introduction to the first soccer lesson, the participants were told that they were to kick a soccer ball with their feet into a goal on the opposing team's court. The same content was used

in each session: 5 minutes of dribbling with a soccer ball as a warm-up, followed by 20 minutes of a soccer game. The teams consisted of four teams of seven to eight players each, and two ten-minute matches were played. The soccer court was 35m long (touchline) x 20m wide (goal line). The soccer goal used was 2m high, 3m wide, and 1m deep. A soccer ball was used in the soccer game, and the rule was that a player could score one point by kicking the ball into the goal of the opposing team's court. If the soccer ball went out of court, one player from the team opposite to the team that kicked the ball out of the court from the place where the ball went out of the court, kicked the soccer ball, and restarted the game. Goalkeepers were not assigned to either team. Children on the team that was not playing the game were instructed to observe the team playing the game. Afterward, the children performed stretch exercises for 5 minutes as a cool-down exercise. Soccer instruction was provided by a male kindergarten teacher who had experience playing soccer and worked at the participants' kindergarten. The decision to participate in soccer instruction was made by the parents before the first soccer instruction session based on their own free will. The intervention group consisted of young children who decided to participate in soccer instruction. Executive function and agility were measured in April 2020 (pre-), before soccer instruction was implemented, and again in August 2020 (post-), after soccer instruction was completed.

While the intervention group received soccer instruction, the control group participated in voluntary indoor play in the kindergarten classroom. The specific content of the indoor play included drawing pictures, making crafts, reading picture books, and playing with toys. The children in the control group did not receive soccer instruction. As with the intervention group, executive function and agility were measured in April 2020 (pre-) and August 2020 (post-).

## Measurements

Executive function and agility were measured by the author, the researcher, in a room at the park where the subject belongs, while the participant was being filmed by a video camera. The results of each measurement were confirmed by reviewing the video recordings. To reduce study bias, the researcher conducting the evaluation was blinded to the group allocation during the evaluation phase.

## Execution functions

The Flanker task [24] was used to measure the inhibition of executive function. Four flanker stimuli consisting of arrow tips (Congruent: < < < < <, > > > > >, Incongruent: > > < > >, < < > < <) were used as the task stimuli. The participants were asked to tap either the left or right option with their fingers according to the orientation of the central target of the flanker stimuli presented on the tablet screen. Four task stimulus patterns were performed in random order with equal probabilities for a total of eight trials. The number of correct responses, number of questions, and elapsed time were used to calculate the performance score (PS) using the formula PS = (number of correct responses/number of questions) × (number of correct responses/elapsed time) [25], with higher PS values indicating greater inhibition ability [25]. This task requires inhibition because the participants must control flanker stimuli other than the central target. The reliability and validity of the flanker task as a measure of inhibition have been confirmed by previous research [24].

To measure working memory, which is part of executive function, we administered the Kaufman Assessment Battery for Children- 2nd Edition (KABC-II) hand movement task [26]. The participants were instructed to memorize and accurately repeat combinations of the evaluator's hand movements in sequence. This task requires visual working memory, which refers

to retaining spatial information and responding quickly to the stored information. One point was awarded for each completed task, and the task was terminated after three consecutive incorrect attempts. Higher scores indicated greater working memory capacity. The reliability and validity of the hand movement task for measuring working memory have been confirmed by previous research [26].

## Agility

A single-line lateral jumping task [27] was performed to measure agility. The participant stood on one side of a single line drawn facing the evaluator. On cue to start, the child performed a side hop repetition with both feet together for 5 seconds. The evaluator counted the number of times the participant jumped over the line and back. A higher number of repetitions indicated greater agility. This task requires agility because the participant must change body position or direction of movement as quickly as possible. The reliability and validity of the single-line lateral jump task in measuring agility have been confirmed in previous studies [27].

## Ethics statement

This study was conducted after explaining the purpose of the study in writing to the parents of the infants who were the participants of the study and obtaining their written consent. We confirmed that they would not be disadvantaged even if they did not consent, that they could withdraw their consent even after providing it and that the personal information obtained in the study would be handled carefully so that it would not be leaked. The recruitment period for participants in this study was from March 23 to March 27, 2020. This study was conducted following the Declaration of Helsinki. This study was approved by the Research Ethics Committee of Hiroshima Jogakuin University, the author's previous institution, on March 14, 2020 (approval number: 2019–12), before the purpose of the study was explained in writing to the participants.

## Data analysis

There was no missing data for the 70 participants in the intervention and control groups combined, so analyses were conducted using data from all subjects. A t-test was used to examine the differences by gender for each measurement item. Cohen's d for t-test was calculated for independent groups. Normality test was conducted for each measurement item to confirm the assumptions of analysis of variance. To examine the effects of the measurement items on development in the intervention and control groups, a two-factor analysis of variance (ANOVA) was conducted for each measurement item in both groups, dividing the items by group (intervention group and control group) and measurement timing (pre- and post-test). All analyses were performed using the IBM SPSS Statistics for Windows version 27 (IBM, Armonk, New York, USA).

## Results

Table 1 shows the results of unpaired t-tests on gender differences in each measurement item. The results showed no significant differences in any of the measures, and no gender differences were observed.

The Shapiro–Wilk test showed that data gathered from each measurement item did not deviate from the normal distribution. Therefore, we conducted a two-factor analysis of variance (ANOVA) for each measurement item by group (intervention group and control group)

**Table 1. Results of unpaired t-tests on gender differences in each measurement item.**

| Measures | pre | | | | t-value | Effect size | post | | | | t-value | Effect size |
|---|---|---|---|---|---|---|---|---|---|---|---|---|
| | Male | | Female | | | | Male | | Female | | | |
| | M | SD | M | SD | t | d | M | SD | M | SD | t | d |
| Flanker task | 0.16 | 0.04 | 0.18 | 0.04 | 1.62 | 0.81 | 0.19 | 0.06 | 0.21 | 0.05 | 1.82 | 0.54 |
| Working memory | 9.46 | 2.52 | 10.40 | 2.60 | 1.54 | 0.37 | 11.54 | 2.23 | 12.31 | 2.19 | 1.46 | 0.35 |
| Agility | 14.94 | 3.66 | 15.26 | 3.51 | 0.37 | 0.09 | 20.57 | 5.57 | 21.29 | 2.98 | 0.67 | 0.16 |

Notes. M = mean; SD = standard deviation

× measurement time (pre and post). The results of the analysis of variance for each measurement item for the participants are shown in Table 2.

Analysis of variance revealed that the interaction was not significant for working memory ($F_{(1,68)} = 0.01$, $p = .92$, $\eta^2 = 0.00$) and agility ($F_{(1,68)} = 0.04$, $p = .84$, $\eta^2 = 0.00$). On the other hand, the interaction for the inhibition was significant ($F_{(1,68)} = 4.86$, $p < .05$, $\eta^2 = 0.07$). According to conventional benchmarks, a small effect size is $\eta^2 = 0.01$, a medium effect size is $\eta^2 = 0.06$, and a large effect size is $\eta^2 = 0.14$ [28]. Thus, the effect size of the interaction for the inhibition was medium to large. A simple main-effect test was conducted because the interaction was significant for the inhibition. As a result, in terms of inhibition, although there was no significant difference in the performance of the intervention group ($M = 0.18$, $SD = 0.01$) and the control group ($M = 0.17$, $SD = 0.01$) at the pre-intervention stage, the intervention group ($M = 0.23$, $SD = 0.01$) showed significantly better performance post-intervention compared to the control group ($M = 0.19$, $SD = 0.01$) ($p < .01$). In terms of inhibition, post ($M = 0.23$, $SD = 0.01$) was significantly higher ($p < .001$) than pre ($M = 0.18$, $SD = 0.01$) in the intervention group, and post ($M = 0.19$, $SD = 0.01$) was significantly higher ($p < .01$) than pre ($M = 0.17$, $SD = 0.01$) in the control group, too.

The results of the analysis of variance for inhibition showed a significant interaction, indicating a difference in the development of the two groups. Although no significant difference between the two groups was observed pre-intervention, the intervention group's post-intervention performance improved significantly more than the control group. However, the results of the analysis of variance for working memory and agility showed no significant interaction. Thus, soccer instruction for kindergarteners with no experience in open-skill sports can significantly improve inhibition but not working memory and agility, where no effect was observed.

## Discussion

This study aimed to investigate whether soccer instruction provided to five-year-old children improved their executive function and agility through comparison with a target group. The

**Table 2. Analysis of variance results for each measurement item.**

| Measures | Intervention | | | | Control | | | | Main effect (time) | Main effect (group) | Interaction | Effect size |
|---|---|---|---|---|---|---|---|---|---|---|---|---|
| | pre | | post | | pre | | post | | F | F | F | $\eta^2$ |
| | M | SD | M | SD | M | SD | M | SD | | | | |
| Flanker task | 0.18 | 0.01 | 0.23 | 0.01 | 0.17 | 0.01 | 0.19 | 0.01 | 33.36*** | 2.24 | 4.86* | 0.07 |
| Working memory | 10.13 | 0.48 | 12.10 | 0.44 | 9.77 | 0.40 | 11.79 | 0.33 | 44.39*** | 0.44 | 0.01 | 0.00 |
| Agility | 15.48 | 0.57 | 21.2 | 0.82 | 14.79 | 0.62 | 20.72 | 0.71 | 123.09*** | 0.50 | 0.04 | 0.00 |

*p < .05
***p < .001

results showed that the intervention group, which received soccer instruction, showed significantly enhanced inhibition performance in executive function compared with the control group. However, there was no difference in the improvement of working memory and agility in the executive function of the intervention group that received soccer instruction compared with the control group.

Several previous studies [12, 14, 15] have reported that players with a higher soccer ability have a higher inhibition ability than those with a lower ability. The present results, in which inhibition was significantly improved by coaching soccer to young children compared with the control group, are similar to those of previous studies. However, the present study revealed that soccer instruction significantly increased inhibition in inexperienced soccer players compared to the control group, considering their ability to inhibit before participating. Soccer requires quick reactions to the movements of the outside world, including the ball, teammates, and opponents. It also requires inhibition because of the need to control one's intended action to select a new and better action among options, such as passing, dribbling, and shooting, in response to constantly changing surroundings [12]. Even beginners who have difficulty handling the ball with their feet and only follow the path of the ball during a soccer game require inhibition to control their behavior according to the path of the ball [12]. Therefore, the inhibition in the intervention group, which received continuous soccer instruction, likely improved to the point where it was significantly higher than that in the control group.

Because soccer requires working memory to constantly update changing information about the surroundings [17], we expected that the development of working memory in the intervention group, which received soccer instruction, would be significantly improved compared to the control group. However, the results of the present study showed no significant differences in the development of working memory between the intervention and control groups. Soccer requires special skills compared to other ball games, mainly in terms of handling the ball with the feet [29]. In addition, feet are less sensitive than hands, and long-term training is required to stop the ball, pass it, and shoot accurately during soccer [29]. When the participants in this study were observed playing a soccer game, some children had difficulty controlling the ball and were desperate to follow its path from start to finish. In this situation, it is difficult for them to update information about their surroundings, such as their opponents and peers, when focusing only on the whereabouts of the ball. Therefore, we suspect that the intervention group, which received soccer instruction, did not show significantly improved development of working memory compared with the control group. Previous studies [13, 15] have reported that the working memory of high-performing soccer players is significantly better than that of controls. However, most studies were conducted on skilled soccer players who had been playing soccer continuously since childhood when they were older than infants. To improve working memory, it may be necessary to provide soccer instruction for a longer period so that the players can control the ball to some extent and thus update information about the movements around them, in addition to paying attention to where the ball is going.

The results showed that soccer instruction was effective in improving the inhibition of executive function in young children but not effective in improving working memory. Executive function develops substantially in the early preschool years. On the other hand, the developmental timing of each component of the executive function is not the same owing to different developmental pathways. In contrast to inhibition, which develops rapidly during infancy, switching and working memory have important developmental periods between the ages of 7 and 9 years [4]. Accordingly, the intervention effects of soccer instruction might have been more pronounced than those of working memory because the five-year-old children who participated in this study were undergoing a critical period in the development of inhibition. A previous study [30] that examined the effects of an intervention program intended to improve

executive function in five-year-old children reported that the effects on inhibition were pronounced, in contrast to working memory, where no intervention effects were observed, showing results similar to those of the present study.

It has been reported that skilled players of open-skill sports, including soccer, have higher agility ability [21, 31]; however, the present results revealed no difference in the development of agility between the intervention and control groups instructed to play soccer. It takes considerable training time for inexperienced soccer players to control the ball and establish a game [32]. Even after 12 weeks of soccer instruction, it was difficult for the inexperienced soccer players to control the ball sufficiently well enough to win a game. In situations where ball control is difficult, it is challenging for young children to demonstrate agility [33], which is required when responding instantaneously to changing external conditions, such as opponents, teammates, and the ball. Therefore, 12 weeks of soccer instruction likely did not improve the agility of inexperienced soccer players compared with the control group. Future research is needed to determine whether a longer period of soccer instruction would improve children's agility to gain control over the ball to compete in a game.

In the present study, inhibition was shown to be more effective than agility for which soccer instruction was ineffective. Inhibition, the ability to contextualize and control dominant, automatic responses, is required when discontinuing intended actions and making new behavioral decisions in response to rapidly changing game situations [1, 14]. Therefore, soccer instructions are expected to improve inhibition. However, even if players can inhibit their intended behaviors, they still need ball control skills to demonstrate agility in executing new behaviors such as passing, dribbling, and shooting. As mentioned earlier, soccer instruction did not lead to improved agility because it was difficult for the participants with no soccer experience to control the ball with their feet, making it challenging for them to demonstrate agility.

Executive function is a cognitive ability whose development can be promoted by intervention and training [34, 35]. Intervention studies intended to improve children's executive function have been conducted worldwide [36]. While it is argued that executive function develops rapidly in early childhood, which is the ideal time to conduct interventions to effectively enhance it [2], few intervention studies have been conducted on young children to improve executive function through exercise. In addition, studies have shown that the executive function of athletes in open-skill sports, particularly soccer, is significantly higher than that of the control group. However, because the original level of executive function before soccer participation cannot be determined, whether soccer participation is effective in improving executive function has not been examined. Regarding these points, the present study revealed that the inhibition of executive function was significantly improved in five-year-old children with no experience of learning open-skill sports, who had never played soccer before, after 12 weeks of continuous soccer instruction compared to the control group. This study is important because it provides new scientific evidence on the value of young children's participation in open-skill sports in terms of the improvement of inhibition of executive function. The transition from kindergarten to elementary school is fraught with various difficulties, and children's inhibition is an essential ability that leads to successful school adjustment, as adjustment to the classroom and learning is a more important issue in elementary school than it was in kindergarten [37]. This study is significant as it demonstrates that participation in open-skill sports can effectively enhance inhibition, which is a critical executive function ability crucial for elementary school readiness.

The results of this study highlight the significance of including open-skill sports such as soccer in the design of physical education programs conducted during nursery hours in the hope of improving inhibition. In recent years, extracurricular activities outside nursery hours, including soccer, have become common worldwide. The results demonstrate the significance

of parents choosing soccer as an extracurricular activity for their children in terms of improving inhibition, which is part of the executive function playing a key role in behavioral and academic readiness for school. A study comparing the executive functions of talented youth soccer players belonging to a youth academy of a professional soccer club with those of amateur youth soccer players reported that although there was no significant difference in working memory, the talented youth soccer players had significantly better inhibition than the amateur youth soccer players [17]. The results of the present study, which targeted kindergarteners with no soccer experience, also suggested that inhibition, one of the executive functions, is particularly required when playing soccer. This result creates new knowledge that inhibition may be essential for success in soccer in various divisions, such as professional and amateur soccer teams.

The present study had several limitations. First, the sample size was relatively small, and the children were from kindergartens in the same area, which limits generalizability. Second, parents choose whether their children would receive soccer instruction as members of an intervention group, which could lead to self-selection bias. Third, this study measured inhibition and working memory, but other aspects of executive function, such as switching, were not evaluated, which limits the results. Future studies are needed to secure a large sample size from a wide range of regions and randomly assign children to the intervention and control groups to examine the effects of participation in open-skill sports on improving executive function and agility.

## Supporting information

**S1 Data.**
(XLSX)

## Acknowledgments

The authors are grateful to the participants for their cooperation.

## Author Contributions

**Conceptualization:** Sho Aoyama.

**Data curation:** Sho Aoyama.

**Formal analysis:** Sho Aoyama.

**Funding acquisition:** Sho Aoyama.

**Investigation:** Sho Aoyama.

**Methodology:** Sho Aoyama.

**Project administration:** Sho Aoyama.

**Resources:** Sho Aoyama.

**Software:** Sho Aoyama.

**Supervision:** Sho Aoyama.

**Validation:** Sho Aoyama.

**Visualization:** Sho Aoyama.

**Writing – original draft:** Sho Aoyama.

**Writing – review & editing:** Sho Aoyama.

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
