## [Decision Letter · Decision Letter 0]

22 Aug 2024

PONE-D-24-26769Effects of soccer instruction on the executive functions and agility of children in early childhoodPLOS ONE

Dear Dr. Aoyama,

Thank you for submitting your manuscript to PLOS ONE. After careful consideration, we feel that it has merit but does not fully meet PLOS ONE’s publication criteria as it currently stands. Therefore, we invite you to submit a revised version of the manuscript that addresses the points raised during the review process.

We look forward to receiving your revised manuscript.

Kind regards,

Mário André da Cunha Espada, PhD

Academic Editor

PLOS ONE

Journal Requirements:

3. Thank you for stating the following financial disclosure: This work was supported by a Grant-in-Aid for Early-Career Scientists Research from the Japan Society for the Promotion of Science [Grant Number 24K20638].

4. Please upload a copy of Supporting Information Figure/Table/etc. S1 which you refer to in your text on page 13.

5. We note you have included a table to which you do not refer in the text of your manuscript. Please ensure that you refer to Table 1 in your text; if accepted, production will need this reference to link the reader to the Table.

Additional Editor Comments:

Dear Authors,

Please consider the recommendations of reviewer 1 and 2 regarding the initial version of the manuscript

Thank you.

Best regards.

Reviewers' comments:

Reviewer's Responses to Questions

**Comments to the Author**

1. Is the manuscript technically sound, and do the data support the conclusions?

Reviewer #1: Yes

Reviewer #2: Yes

2. Has the statistical analysis been performed appropriately and rigorously? 

Reviewer #1: Yes

Reviewer #2: I Don't Know

3. Have the authors made all data underlying the findings in their manuscript fully available?

Reviewer #1: Yes

Reviewer #2: Yes

4. Is the manuscript presented in an intelligible fashion and written in standard English?

Reviewer #1: Yes

Reviewer #2: No

5. Review Comments to the Author

Reviewer #1: Overall, the study is an interesting idea but for better readability it needs improvements in writing and improve particularly in the study's methodology and results that will be discussed in the relevant section.

Abstract:

The abstract could be better structured to enhance readability.

The abstract does not mention the outcome related to agility, even though it is a central focus of the study.

In the results sentences: Clarify the comparison of pre- and post-test scores within the intervention and control groups, as the current phrasing might confuse readers.

In overall, by improving the clarity, structure, and inclusion of all relevant results, the abstract would be more comprehensive and aligned with the standards of the journal.

Introduction:

The introduction provides a comprehensive overview of the relevance of exercise, particularly open-skill sports like soccer, in enhancing executive function and agility in children. Meanwhile, there are some comments for improving clarity and the flow:

The introduction repeats some information, particularly regarding inhibition and the advantages of open-skill sports. Condensing these points would improve readability, e.g., the paragraph discussing the benefits of open-skill sports (line 86-97) could be streamlined by merging similar ideas and avoiding redundancy in discussing inhibition tasks across different studies.

The section discussing the limitations of previous studies in establishing a causal relationship between soccer participation and improved executive function could be more precise (line:64-70 and 79-85). It might help to explicitly state that the lack of baseline measures is a significant limitation in previous research.

The purpose of the study is clear, but it could be emphasized more by separating it into its own paragraph and explicitly stating the research questions or hypotheses.

Method:

To report average and mean of the age for five years old, by months or year ±SD, Refer to the PLOS one priority format (line 110- 114).

The methodology is well-structured and covers essential aspects of the study, but it has several weaknesses that could affect the study's validity, reliability, and generalizability. Addressing these issues would support the study design and improve the strength of the findings, e.g.,:

As a Small Sample Size study: The intervention group consists of only 31 children, and the control group has 39. While this might be sufficient for a pilot study, the small sample size limits the generalizability of the findings. The results may not be representative of the broader population of five-year-old children. Then it needs to be considered as the study limitation.

As a Single Study Location: All participants are from the same kindergarten in the same prefecture, which further limits generalizability. The study’s findings might not apply to children in different geographical locations, cultural settings, or educational environments. If there is not any confidential conflict, it is better to write fully name of kindergarten (line 110 and 113).

It needs to clarify the randomization. Since the decision to participate in soccer instruction was made by the parents, leading to a potential self-selection bias. Then, a randomized controlled trial (RCT) design would be stronger, if children were randomly assigned to either the intervention or control group to minimize selection bias.

To control group activity, it needs to provide a detailed description of the control group’s activities or consider using an active control group that participates in another structured, non-soccer activity.

Blinding the researcher to group allocation during the assessment and analysis phases would support lessen the study bias.

Outcome Measures: The study measures inhibition and working memory, but other aspects of executive function, such as cognitive flexibility and planning, are not assessed that can be considered as limitation of the study findings.

Statistical Analysis: Lack of Detail on Statistical Methods: Include a more detailed description of the statistical methods used, such as the type of tests (e.g., t-tests, ANOVA, regression analysis), how assumptions were checked, and how missing data were handled. While the methodology mentions using a two-way ANOVA, it’s important to ensure that the assumptions of ANOVA (e.g., homogeneity of variances, normality) are met.

No Mention of Effect Size or Power Analysis: The section does not mention whether effect sizes were calculated or whether a power analysis was conducted to determine the appropriate sample size.

Ethic: Include the name of the institution, the approval number, and the date of approval.

Results:

The results section provides a solid foundation for presenting the study's findings, but there are several areas where clarity and detail could be improved.

Clarify the interpretation of the interaction effect. Explain what the significant interaction means in the context of the study. For example, discuss whether the intervention was more effective for certain groups or conditions and how this relates to the overall study objectives.

Provide a brief interpretation of the effect sizes, explaining whether they are small, medium, or large according to conventional benchmarks. This will help readers understand the practical significance of the findings. Ensure that the comparison between pre and post-intervention within the control group is clearly stated, without unnecessary repetition or ambiguity

Discussion

Discussion is well-structured and provides a comprehensive analysis of the study's findings. However, I have a few suggestions to improve clarity, coherence, and flow:

The discussion touches on the practical implications of the findings, particularly for early childhood education, but these points could be more clearly articulated.

Improvement: Expanding on how these findings could influence the design of physical education programs, or how educators and parents might apply these insights, would strengthen the practical relevance of the study.

Reviewer #2: Abstract (line 30-32): is this correct” In the intervention group, the “post-” score was significantly higher than the “pre-” score, whereas in the control group, the “post-” score was significantly higher than of the “pre-” score. Please reword.

Study needs a hypothesis.

Did participants in the intervention group have any previous experience in playing soccer---this is briefly mention in the Discussion (line 224)? Were they naïve to soccer playing until this study started? Was this performed on a regular sized soccer field? What about the size of the goal posts—same as in adult soccer?

Unclear if both or only the control group played indoor soccer?

The description of the statistics used is very limited. What is “T-value”? How was effect size calculated? The results section should better describe the two groups in terms of their characteristics. What were the inclusion / exclusion criteria sued when recruiting participants?

What are the implications of these findings for professional vs amateur teams, and also for different divisions in professional leagues?

6. PLOS authors have the option to publish the peer review history of their article (what does this mean?). If published, this will include your full peer review and any attached files.

Reviewer #1: **Yes: **Fariba Hossein Abadi

Reviewer #2: No

---

## [Author Response · Author response to Decision Letter 0]

5 Sep 2024

Rebuttal Letter

Dear Editor,

Thank you for the opportunity to revise my manuscript entitled “Effects of soccer instruction on the executive functions and agility of children in early childhood” (PONE-D-24-26769). In revised manuscript, I have carefully considered reviewers’ comments and suggestions. As instructed, I have attempted to succinctly explain changes made in reaction to all comments and replied to each comment point-by-point, and those responses are provided below. I have color-coded the appropriate text in the revised manuscript. The reviewers’ comments were very helpful in revising the manuscript, and I appreciate receiving such constructive feedback on my original submission. After addressing the issues raised, the quality of the paper is much improved.

Sincerely

Sho Aoyama, PhD

A. Thank you for your suggestion. I have reviewed the revised manuscript to ensure that it meets PLOS ONE style requirements.

A. The ORCID iD obtained has been verified in Editorial Manager.

3. Thank you for stating the following financial disclosure: This work was supported by a Grant-in-Aid for Early-Career Scientists Research from the Japan Society for the Promotion of Science [Grant Number 24K20638].

A. The cover letter (revised) has been updated to include the following statement: The funders had no role in study design, data collection and analysis, decision to publish, or preparation of the manuscript.

4. Please upload a copy of Supporting Information Figure/Table/etc. S1 which you refer to in your text on page 13.

A. The raw data required to replicate the results of the present study created in Excel with the file name 'S1' is uploaded as Supporting Information in the Attach Files.

5. We note you have included a table to which you do not refer in the text of your manuscript. Please ensure that you refer to Table 1 in your text; if accepted, production will need this reference to link the reader to the Table.

A. The reference to Table 1 has been added to the main text (lines 216–217).

Reviewer: 1

Thank you for the time to review this manuscript for improvement. I have revised the manuscript based on your comments, and I believe they have all been thoroughly addressed. In the file titled “Revised Manuscript with Track Changes,” the revised text is highlighted in yellow. I have included my responses to your comments below.

Abstract:

The abstract could be better structured to enhance readability. The abstract does not mention the outcome related to agility, even though it is a central focus of the study. In the results sentences: Clarify the comparison of pre- and post-test scores within the intervention and control groups, as the current phrasing might confuse readers. In overall, by improving the clarity, structure, and inclusion of all relevant results, the abstract would be more comprehensive and aligned with the standards of the journal.

A: Thank you for your suggestions. I added the results of the ANOVA for agility and working memory to the abstract. I also clarified the comparison of pre-test and post-test scores within the intervention and control groups. I improved the clarity of the abstract by revising it to clearly express the limitations from previous studies, the purpose of this study, and the main results of this study.

Introduction:

The introduction provides a comprehensive overview of the relevance of exercise, particularly open-skill sports like soccer, in enhancing executive function and agility in children. Meanwhile, there are some comments for improving clarity and the flow:

The introduction repeats some information, particularly regarding inhibition and the advantages of open-skill sports. Condensing these points would improve readability, e.g., the paragraph discussing the benefits of open-skill sports (line 86-97) could be streamlined by merging similar ideas and avoiding redundancy in discussing inhibition tasks across different studies.

The section discussing the limitations of previous studies in establishing a causal relationship between soccer participation and improved executive function could be more precise (line:64-70 and 79-85). It might help to explicitly state that the lack of baseline measures is a significant limitation in previous research.

The purpose of the study is clear, but it could be emphasized more by separating it into its own paragraph and explicitly stating the research questions or hypotheses.

A: Thank you for sharing your concerns. In the introduction section, I have revised the article to avoid redundancy by summarizing previous studies which show that elite open skill sports athletes have significantly higher inhibition than control groups (lines 61–69). 

I have also revised the article to clearly state that the lack of baseline measurements was a challenge in previous studies in establishing a causal relationship between soccer participation and improved executive function (lines 73–77). 

I have revised the article to separate the purpose of this study into its own paragraph and explicitly state the research questions and hypotheses (lines 98–102 and 105–107).

Method:

To report average and mean of the age for five years old, by months or year ±SD, Refer to the PLOS one priority format (line 110- 114).

The methodology is well-structured and covers essential aspects of the study, but it has several weaknesses that could affect the study's validity, reliability, and generalizability. Addressing these issues would support the study design and improve the strength of the findings, e.g.,:

As a Small Sample Size study: The intervention group consists of only 31 children, and the control group has 39. While this might be sufficient for a pilot study, the small sample size limits the generalizability of the findings. The results may not be representative of the broader population of five-year-old children. Then it needs to be considered as the study limitation.

As a Single Study Location: All participants are from the same kindergarten in the same prefecture, which further limits generalizability. The study’s findings might not apply to children in different geographical locations, cultural settings, or educational environments. If there is not any confidential conflict, it is better to write fully name of kindergarten (line 110 and 113).

It needs to clarify the randomization. Since the decision to participate in soccer instruction was made by the parents, leading to a potential self-selection bias. Then, a randomized controlled trial (RCT) design would be stronger, if children were randomly assigned to either the intervention or control group to minimize selection bias.

To control group activity, it needs to provide a detailed description of the control group’s activities or consider using an active control group that participates in another structured, non-soccer activity.

Blinding the researcher to group allocation during the assessment and analysis phases would support lessen the study bias.

Outcome Measures: The study measures inhibition and working memory, but other aspects of executive function, such as cognitive flexibility and planning, are not assessed that can be considered as limitation of the study findings.

Statistical Analysis: Lack of Detail on Statistical Methods: Include a more detailed description of the statistical methods used, such as the type of tests (e.g., t-tests, ANOVA, regression analysis), how assumptions were checked, and how missing data were handled. While the methodology mentions using a two-way ANOVA, it’s important to ensure that the assumptions of ANOVA (e.g., homogeneity of variances, normality) are met.

No Mention of Effect Size or Power Analysis: The section does not mention whether effect sizes were calculated or whether a power analysis was conducted to determine the appropriate sample size.

Ethic: Include the name of the institution, the approval number, and the date of approval.

A: Thank you for your suggestion. In the method section, the mean and standard deviation of participants’ ages in months were corrected to be stated in years ± SD (lines 114–117). 

As you pointed out, the relatively small sample size, participant recruitment from the same region, and the parental decision (rather than the participant’s decision) to participate in soccer instruction, which may have led to self-selection bias, are limitations of this study that need to be considered. I have added these as limitations to the Discussion section (lines 376–379). Unfortunately, confidentiality requirements prevent me from sharing the full name of the kindergarten.

To control for group activities, I added a detailed description of the control group's activities (lines 148–151).

I have added that, to reduce study bias, researchers were blinded to group allocation during the evaluation phase (lines 158–159).

As you point out, this study was unable to assess other aspects of executive function, such as switching, so I have added this as a limitation in the Discussion section (lines 379–381).

I added details about the types of tests, how assumptions were checked, how missing data were handled, and statistical methods, such as effect sizes and power analyses (lines 121–124, 203–213, and 223–224).

I have added the name of the institution that conducted the ethical review for this study, the approval number, and the approval date to the text (lines 199–202).

Results:

The results section provides a solid foundation for presenting the study's findings, but there are several areas where clarity and detail could be improved.

Clarify the interpretation of the interaction effect. Explain what the significant interaction means in the context of the study. For example, discuss whether the intervention was more effective for certain groups or conditions and how this relates to the overall study objectives.

Provide a brief interpretation of the effect sizes, explaining whether they are small, medium, or large according to conventional benchmarks. This will help readers understand the practical significance of the findings. Ensure that the comparison between pre and post-intervention within the control group is clearly stated, without unnecessary repetition or ambiguity.

A: Thank you for sharing your concerns. I have added an explanation of the interaction effect to clarify its meaning considering the purpose of this study (lines 238–254). I have also added an explanation of the effect size (lines 235–237).

 After clearly explaining the main results of the ANOVA on inhibition, I revised the section to report only the results of the pre–post comparisons for the intervention and control groups, respectively (lines 243–246).

Discussion

Discussion is well-structured and provides a comprehensive analysis of the study's findings. However, I have a few suggestions to improve clarity, coherence, and flow:

The discussion touches on the practical implications of the findings, particularly for early childhood education, but these points could be more clearly articulated.

Improvement: Expanding on how these findings could influence the design of physical education programs, or how educators and parents might apply these insights, would strengthen the practical relevance of the study.

A: Thank you for sharing your concerns. In the discussion section, I have added a clear explanation of how the results of this study can be applied to teachers and parents, including the design of physical education programs (lines 360–366).

Reviewer: 2

Thank you for the time to review this manuscript. I have completed revisions based on your comments, and I believe they have been fully addressed. In the file titled “Revised Manuscript with Track Changes,” the revised text is highlighted in yellow. Responses to each of your comments are included below.

Abstract (line 30-32): is this correct” In the intervention group, the “post-” score was significantly higher than the “pre-” score, whereas in the control group, the “post-” score was significantly higher than of the “pre-” score. Please reword.

A: Thank you for your suggestion. I have revised the wording in the abstract as follows to avoid misunderstandings among readers regarding the results of the analysis of variance on inhibition as the main result of this study.

‘Regarding inhibition, although the intervention group (M = 0.18, SD = 0.01) and the control group (M = 0.17, SD = 0.01) did not differ significantly in performance pre-intervention, the intervention group (M = 0.23, SD = 0.01) showed significantly better performance post-intervention compared to the control group (M = 0.19, SD = 0.01) (p < .01).’

Study needs a hypothesis.

A: Thank you for sharing your concerns. I have added the hypotheses of this study in the introduction section (lines 105–107).

Did participants in the intervention group have any previous experience in playing soccer---this is briefly mention in the Discussion (line 224)? Were they naïve to soccer playing until this study started? Was this performed on a regular sized soccer field? What about the size of the goal posts—same as in adult soccer?

A. Thank you for your suggestion. In the methods section, I noted that the study participants had no prior participation in open skill sports, including soccer (lines 118–120). The soccer played by the intervention group is different from soccer played by adults. I added details about the size of the court and goal size on which the intervention group played soccer (lines 132–134).

Unclear if both or only the control group played indoor soccer?

A. Thank you for noting this issue. Only the intervention group received soccer instruction; I have added a note to the methods section indicating that the children in the control group did not receive soccer instruction (line 151).

The description of the statistics used is very limited. What is “T-value”? How was effect size calculated? The results section should better describe the two groups in terms of their characteristics. What were the inclusion / exclusion criteria sued when recruiting participants?

A. Thank you for highlighting your concerns. I have added explanations of the statistics used, including t-tests, analysis of variance, missing data, how assumptions were checked, and calculations of effect sizes in the data analysis section (lines 121–122 and 203–213). 

 In the results section, I have added a description of the characteristics of the two groups regarding inhibition for which the interaction was significant (lines 238–246).

 In the methods section, I have added a description of the exclusion criteria used to recruit participants who had never participated in soccer, including open skill sports (lines 118–120).

What are the implications of these findings for professional vs amateur teams, and also for different divisions in professional leagues?

A. In the discussion section, I have added information about the impact the results of this study have on various sectors, includi

---

## [Decision Letter · Decision Letter 1]

4 Oct 2024

Effects of soccer instruction on the executive functions and agility of children in early childhood

PONE-D-24-26769R1

Dear Dr. Sho Aoyama,

We’re pleased to inform you that your manuscript has been judged scientifically suitable for publication and will be formally accepted for publication once it meets all outstanding technical requirements.

Kind regards,

Mario André da Cunha Espada, PhD

Academic Editor

PLOS ONE

Additional Editor Comments:

Dear Authors,

Congratulations on the work carried out in the review process.

I propose that the authors consider the suggestions in this last phase of revision (e.g. English details improvement).

Thank you.

Best regards.

Reviewers' comments:

Reviewer's Responses to Questions

**Comments to the Author**

1. If the authors have adequately addressed your comments raised in a previous round of review and you feel that this manuscript is now acceptable for publication, you may indicate that here to bypass the “Comments to the Author” section, enter your conflict of interest statement in the “Confidential to Editor” section, and submit your "Accept" recommendation.

Reviewer #2: All comments have been addressed

Reviewer #3: All comments have been addressed

2. Is the manuscript technically sound, and do the data support the conclusions?

Reviewer #2: Yes

Reviewer #3: Yes

3. Has the statistical analysis been performed appropriately and rigorously? 

Reviewer #2: Yes

Reviewer #3: Yes

4. Have the authors made all data underlying the findings in their manuscript fully available?

Reviewer #2: Yes

Reviewer #3: Yes

5. Is the manuscript presented in an intelligible fashion and written in standard English?

Reviewer #2: No

Reviewer #3: Yes

6. Review Comments to the Author

Reviewer #2: Please edit the manuscript for English language and grammar. Perhaps recruit assistance from a native English language speaker

Reviewer #3: Thank you for the opportunity to review the manuscript with the title - Effects of soccer instruction on the executive functions and agility of children in early childhood.

The article is well structured, and the improvements made are opportune and contribute to increasing the scientific value of the manuscript.

Recommendations: - The article is well structured and meets the scientific requirements of the journal in order to be accepted.

7. PLOS authors have the option to publish the peer review history of their article (what does this mean?). If published, this will include your full peer review and any attached files.

Reviewer #2: **Yes: **Ismail Laher

Reviewer #3: No

---

## [Editor Report · Acceptance letter]

14 Oct 2024

PONE-D-24-26769R1 

PLOS ONE

Dear Dr. Aoyama, 

I'm pleased to inform you that your manuscript has been deemed suitable for publication in PLOS ONE. Congratulations! Your manuscript is now being handed over to our production team.

Kind regards, 

on behalf of

Dr. Mario André da Cunha Espada 

Academic Editor

PLOS ONE